# Application of Indocyanine Green in Combination with Da Vinci Xi Robot in Surgeries on the Upper Urinary Tract: A Case Series Study

**DOI:** 10.3390/jcm12051980

**Published:** 2023-03-02

**Authors:** Sheng Zeng, Shaoqiang Xing, Wenzhou Xing, Zhijie Bai, Jingyuan Zhang, Yanan Li, Haifeng Wang, Qian Liu

**Affiliations:** 1Department of Urology, Tianjin First Central Hospital, Tianjin 300192, China; 2The First Clinical Medical College, Tianjin Medical University, Tianjin 300192, China; 3Department of Operating Room, Tianjin First Central Hospital, Tianjin 300192, China

**Keywords:** Da Vinci robot, indocyanine green, collecting system of the urinary tract, upper urinary tract, case-series system

## Abstract

Background: To explore the application value of intraoperative imaging by indocyanine green (ICG) injection through the collection system of the urinary tract for Da Vinci Xi robot navigation in complex surgeries on the upper urinary tract. Methods: Data of 14 patients who underwent complex surgeries of the upper urinary tract post-ICG injection through the collection system of the urinary tract in combination with Da Vinci Xi robot navigation in the Tianjin First Central Hospital between December 2019 and October 2021 were analyzed in this retrospective study. The operation duration, estimated blood loss, and exposure time of ureteral stricture to ICG were evaluated. The renal functions and tumor relapse were evaluated after surgery. Results: Of the fourteen patients, three had distal ureteral stricture, five had ureteropelvic junction obstruction, four presented duplicate kidney and ureter, one had a giant ureter, and one presented an ipsilateral native ureteral tumor after renal transplantation. The surgeries in all patients were successful, with no conversion to open surgery. In addition, no injury to the surrounding organs, anastomotic stenosis or leakage, or ICG injection-related side effects were detected. Imaging at 3 months post-operatively revealed improved renal functions compared to those before the operation. No tumor recurrence or metastasis was observed in patient 14. Conclusion: Fluorescence imaging compensating for the inadequacy of tactile feedback in the surgical operating system has advantages in identifying the ureter, determining the site of ureteral stricture, and protecting the blood flow for the ureter.

## 1. Introduction

The advancement of endoscopy techniques in recent years has increased the number of ureteroscopy-induced iatrogenic injuries. In addition, inflammation, retroperitoneal fibrosis, and changes after renal transplantation procedures induced ureteral ischemic obstruction [1,2]. Complex ureteral repair and reconstruction are a substantial challenge in urinary surgeries. Ureterectomy includes the resection of ureteral segments with obstruction and/or scars and protection of healthy ureteral segments, which requires tensionless and water-proof anastomosis with good vascularization. However, the poor tissue plane, fibrosis, and anatomical abnormalities complicate the surgeries.

The Da Vinci robot has been used extensively in complex surgeries of the upper urinary tract in the urinary system due to its advantages, such as flexible operation and sophisticated and high-resolution three-dimensional images. Compared to open surgeries, surgeries with the Da Vinci robot involve smaller wounds and a faster recovery, however, there are some difficulties in rapidly and accurately identifying the site of the lesion [3]. Indocyanine green (ICG) is a non-radiative fluorescent contrast agent. In recent years, intraoperative ICG fluorescence imaging in combination with Da Vinci robot has shown unique advantages in urological surgical procedures [4,5,6]. Previous studies have reported satisfactory clinical effects of ICG in laparoscopic surgeries for ureteral repair [6]. This technique could assist urologists in accurately identifying the stricture segment of the ureter; however, only a few studies reported the application of this technique in complex upper urinary tract repair surgeries. According to the different surgical requirements and sites of surgeries, the application of ICG could be categorized into peripheral venous injection [4], para-tumor injection [7], and injection to the collection system of the urinary tract [8]. In this study, ICG was injected through the collecting system of the urinary tract (through the ureteral catheter or nephrostomy tube) to achieve intraoperative fluorescence imaging, in combination with Da Vinci Xi robot navigation for complex surgeries of the upper urinary system. Herein, we speculated that ICG injection through the collecting system of the urinary tract achieves visualization under fluorescence and assists in upper urinary tract surgeries. This present study aimed to report the findings and explore the feasibility of the application of ICG using Da Vinci Xi robot in complex upper urinary system surgeries.

## 2. Materials and Methods

### 2.1. Study Design and Patients

Data from 14 patients scheduled for upper urinary tract surgeries, according to their disease conditions, in the Tianjin First Central Hospital between December 2019 and October 2021 were retrospectively analyzed. None of the patients had undergone upper urinary tract surgery previously. All the patients understood the adverse responses of ICG injection administered through the collection system of the urinary tract and consented to the injection and visualization of the Da Vinci robot surgery under near-infrared fluorescence. The 14 patients included 6 males and 8 females, and the mean age of the cohort was 44.3 ± 8.7 years. Computed tomography (CT) and renal nuclide scanning before the operation evaluated the disease and renal functions of all 14 patients. The operative time, estimated blood loss, and exposure time of ureteral stricture under ICG were recorded. Imaging evaluation of uronephrosis, tumor relapse, and renal functions was conducted 3 months after surgery.

Of the 14 patients, 3 had distal ureteral stricture, 5 had ureteropelvic junction obstruction, 4 had duplicate kidney and duplicate ureter, 1 presented a megaureter, and 1 had ipsilateral native ureteral tumor after renal transplantation. All patients underwent antegrade imaging and CT examination of the urinary system (Table 1). Of the 3 patients with distal ureteral stricture, 2 had the stricture on the left side (Figure 1A); all the 3 patients had moderate or severe uronephrosis, and the ipsilateral renal function was 23.2 ± 2.15 mL/min. Of the 5 patients with ureteropelvic junction obstruction, the stricture was on the left in 3 patients; all 5 patients had moderate or severe uronephrosis, and the ipsilateral renal function was 28.5 + 5.2 mL/min. The ipsilateral renal function of 4 patients with a duplicate kidney and duplicate ureter was 36.3 ± 4.6 mL/min. One of the patients presented stricture of the ureteropelvic junction obstruction and multiple calculi of the lower renal cortex that had a normal thickness. The patient was treated by percutaneous nephrolithotomy, and a nephrostomy tube was implanted (Figure 1B). Another patient had a duplicate ureter with an opening at the bladder neck, urinary incontinence, evident ureterectasia and uronephrosis, and thinning of the cortex at the duplicated renal segment. Full-length ureterectasia and severe uronephrosis were detected in the patient with a megaureter, and the split renal function was 17.5 mL/min (Figure 1C). In the case of a patient with an ipsilateral native ureteral tumor after renal transplantation, the tumor was at the lower segment of the ureter, and the native ureter adhered to the renal artery of the transplanted kidney (Figure 1D). This study was approved by the Ethics Committee of the hospital, and all patients signed the informed consent.

### 2.2. Selection of Surgical Procedures

#### 2.2.1. Preoperative Preparation

For patients with hydronephrosis accompanied by ureteral stricture, the nephrostomy tube was implanted 7 days before the surgery. In patients with duplicate kidneys and ureters accompanied by lower renal calculi, percutaneous nephrolithotomy was performed 3 months before the operation, and the nephrostomy tube was implanted. After the fistulation tube was clamped, lower hydronephrosis was found, and the ureteral catheter was placed at the upper ureter. For all the other patients, ureteral catheters were implanted under cystoscopy. All the surgeries were performed by Professor Liu et al. The fluorescence imaging system of the Da Vinci robot was used in operation for imaging of the surgery region.

#### 2.2.2. ICG Preparation and Administration

In this study, 25 mg ICG was solubilized in 50 mL sterile water for injection. The patients were placed in a surgical position, the ureter was separated and exposed by the operators, and then the ICG was injected into the renal pelvis or ureter through the nephrostomy tube or ureteral catheter. Subsequently, the fluorescence imaging system of the Da Vinci robot was used for imaging navigation. Nausea, fever, shock, and other side effects may occur after ICG enters the blood system.

#### 2.2.3. Selection of the Surgical Mode on Upper Urinary Tract

Upper urinary tract reconstruction: The ureter was distinguished by observing the fluorescence images, and the segment of ureteral stricture was identified by the fluorescence difference. Typically, the ureteral segment with weak or no fluorescence was the site of the stricture. For patients with distal ureteral stricture, the ureter was separated and resected at the site of accessing bladder and then anastomosed to the bladder (Figure 2A,B). All the patients underwent ureterovesical anti-reflux anastomosis, with the anastomosis site at the base of the bladder. For patients with ureteropelvic junction obstruction, renal pelvis-ureter “Y-V plasty” was performed, following which the stricture segment was resected, D-J tube was implanted, and the renal pelvis was sutured to the ureter. During the procedures, the blood flow for the ureter was protected. Resection was performed for patients with duplicate kidneys and ureters. For patient 12, percutaneous nephrolithotomy was performed, and then fluorescence imaging was used to identify the segment of ureteral stricture, the site of ureter resection, followed by end-to-side anastomosis of the ureter with lower renal pelvis was performed (Figure 2C,D). For the patient with a megaureter, distal ureter plication followed by ureteroneocystostomy was performed.

Native ureterectomy of transplanted kidney: The renal artery of the transplanted kidney was closely adjacent to the primary ureter in the patient with native ureteral tumor. The native ureter was observed by imaging to identify the locations of the ureter and the artery, thus avoiding accidental damage to the renal artery during the operation, then, the primary kidney, ureter, and part of the bladder were resected (Figure 2E,F).

Assessment of anastomotic stenosis or leakage after reconstruction: After anastomosis of the ureter was completed, ICG was injected again, and ICG leakage or accumulation at anastomosis was evaluated to assess the presence of anastomotic stenosis or leakage.

## 3. Results

The ICG solution was injected through the ipsilateral ureteral catheter or nephrostomy tube and flowed into the ureter. The Da Vinci robot was guided by the ICG fluorescence, and the surgeries were performed successfully in all 14 patients. Patients 1, 2, and 3 with distal ureteral stricture underwent ureterovesical anti-reflux anastomosis. Patients 4, 5, 6, 7, and 8 with ureteropelvic junction obstruction underwent pyeloplasty. Patients 9, 10, and 11 with duplicate kidneys and ureters underwent resection of the duplicate kidney and ureter. Patient 12 underwent percutaneous nephrolithotomy + end-to-side anastomosis of the ureter with the lower renal pelvis. Patient 13 with a megaureter underwent resection of the stricture segment of the distal ureter, followed by distal ureter application and ureteroneocystostomy. Patient 14 with a native ureteral tumor after renal transplantation underwent resection of the native kidney, native ureter, and partial bladder. No conversion to open surgery was required for the 14 patients; the operative time was 143.2 ± 23.1 min, estimated blood loss was 74.3 ± 29.7 mL, and the duration required for separating the ureter was 23 ± 6.6 min. No urinary leakage or stenosis was observed at the site of ureteral anastomosis.

The surgeries were successful in all 14 patients, and there was no conversion to open surgery. The operation time was 143.2 ± 23.1 min, the estimated blood loss was 74.3 ± 29.7 mL, and the duration for ureter separation was 23 ± 6.6 min. No urinary leakage or stenosis occurred at the site of ureteral anastomosis. The mean length of the hospital stay was 8 days for the patients. No injury of the renal artery was detected in the transplanted kidney in patient 14. No ICG-related side effect was observed in the patients. Then, the D-J and nephrostomy tubes were successfully removed 3 months after the operation, and color ultrasound examination of the urinary system did not show uronephrosis or ureterectasia. The ipsilateral renal function did not worsen further, and the mean renal function was 37.9 ± 6.5 mL/min. Abdominal CT scanning of patient 14 did not show and tumor recurrence. All 14 patients achieved clinical and imaging success (Table 2).

## 4. Discussion

Ureterectomy includes the resection of ureteral segments with obstruction and/or scars and protection of healthy ureteral segments, which requires tensionless, water-proof anastomosis with good vascularization. However, the poor tissue plane, fibrosis, and anatomical abnormalities lead complicate the surgeries. Imaging-guided surgery could improve the simplicity, accuracy, and comfort of complex procedures. For instance, X-ray images could be used to guide the surgeries on the ureter, bladder, urethra, and ureteropelvic junction [9]. However, images are not capable of guiding real-time surgery, and cannot reflect the intraoperative conditions, which further limits the use of imaging guidance. Laparoscopic surgery can not accurately display the tumor boundary or differentiate the surgical part from the surrounding tissue boundary under ordinary light. ICG is a real-time contrast agent with high tissue permeability, high signal-to-noise ratio, and extremely high safety, deeming it a promising agent for optic imaging to improve the visualization of interior anatomical structures and is suitable intraoperative. During surgery, indocyanine green can be used to stimulate intraoperative fluorescence, which can better enable negative tumor imaging or identify tissue boundaries The combination of local imaging and minimally invasive surgery is a significant advancement in surgical approaches. Previous studies reported that in urinary system-related surgeries, ICG was first used in sentinel lymph node dissection in radical resection of bladder and prostate cancers [10,11], distinguishing normal renal parenchyma and tumor tissues in partial nephrectomy, as well as selective renal arterial clamping [12] and positioning of tumor in partial adrenalectomy [13]. Jan et al. [14] reported that in lymph node dissection by Da Vinci robot, a mixed fluorescence ICG radioactive tracer was used to detect the sentinel lymph nodes, and the intraoperative visualization percentage was improved to 93.5% by fluorescence imaging, while that in the control group was 50.0%. Borofsky et al. [6] described ICG guidance for super-selective renal arterial clamping in 27 patients who underwent partial nephrectomy, which prevented injuries to healthy renal parenchyma. The highest loss of EGFR at 3-month follow-up was 1.6% in the ICG group and 14.0% in the non-ICG group. These findings demonstrated that using ICG in surgeries could increase the treatment effects and reduce collateral injuries.

Due to limited operating space, the small opening, and poor blood flow for the ureter, the procedures for ureteral reconstruction are challenging. Although the identification of the ureter and stenosis segment is a minor step of the overall surgery, it is most challenging in the procedures of ureteral reconstruction that could influence the outcomes of the surgery. Typically, such reconstruction procedures are performed in open surgeries. Although various studies have reported that this reconstruction is applicable in the ureteral reimplantation under laparoscopy [15], these surgeries have not been extensively applied compared to open surgeries due to pelvic anatomy and technical difficulties of in-body suturing [16,17]. The application of robot technique endowed complex urinary surgeries with the advantages of high precision, miniaturization of devices, and small incisions compared to laparoscopic or open surgeries. The Da Vinci robot provided a three-dimensional visualization, the “wrist” of the robot is more flexible than the laparoscope, and the robot is capable of operating from different angles around the target organs. In addition, the robot does not fatigue, and the skills of suturing and knotting are excellent [18]. The Da Vinci Xi robot is equipped with a fluorescence imaging system, and the combination with ICG fluorescence imaging could make image-guided surgery more convenient. In addition, using fluorescence imaging to compensate for the inadequacy of tactile feedback in the surgical operating system reduces the damage to the surrounding tissues. ICG in combination with the Da Vinci robot simplifies the procedures of ureteral reconstruction, including the procedures of ureterolysis, ureteral anastomosis, ureteral reimplantation, and pyeloplasty [19]. Bjurlin et al. [20] reported intravenous injection of ICG in urinary reconstruction in 43 patients, which assists in the identification of the ureter and positioning of the stricture segment of the ureter. According to the findings, ICG displays the whole surgical region, the blood supply for normal ureter, and the fluorescence intensity were high, while the stricture segment had poor blood flow and the fluorescence intensity was low. However, the influence of background fluorescence was substantial. The active ureteral mucosal cells have a high protein content and can easily bind to ICG, directly injecting ICG into the collecting system of the urinary system (i.e., into the ureter or renal pelvis) for fluorescence imaging and easy positioning of the ureter and the segment of stricture [21]. Lee et al. [22] injected ICG through the ureteral catheter or nephrostomy tube in seven patients who underwent robot-assisted ureteral anastomosis; no ICG-related complication was detected during or after the operation. The ICG injection through the collecting system of the urinary tract distinguishes between the healthy ureter and lesion tissues and further assists in robot-assisted ureteral repair. In this study, ICG was injected through the collection system of the urinary tract in all 14 patients, the background inference was reduced in imaging, the fluorescence contrast was evident in the surgery region, the stricture site of the ureter was identified rapidly, and the surrounding normal tissues could be well-distinguished.

Notably, for patients with distal ureteral stricture, the ureter was separated and resected at the site with weak or no fluorescence; then D-J tube was implanted, which was connected to the distal ureter by a 7-0 thread, and the ureter was anastomosed to the bladder. The re-examination 3 months after the operation in three patients with distal ureteral stricture showed that the ipsilateral uronephrosis alleviated and renal function improved. For patients with a megaureter, the segment with functional obstruction was resected at the site with weak fluorescence, and gauffer suturing was essential at the exterior wall of the ureter 5 cm from the distal end, which reduced the lumen of the megaureter [23]. Importantly, F6 ureteral catheter could be used for scaffold drainage, after which the ureter could be anastomosed to the bladder. For patients with ureteropelvic junction obstruction, pyeloplasty is influenced by high anastomotic tension, with difficulty and prolonged anastomosis. For such patients, the stricture segment was incompletely resected, and then the distal ureter was vertically cut open, which passed through the stricture segment and further cut downward for approximately 1–1.5 cm. The proximal renal pelvis wall was cut open in a “V” shape, and then the renal pelvis-ureter “Y-V plasty” was performed [24,25]. Subsequently, the segment of the stricture was completely resected, the D-J tube was implanted, and the renal pelvis and ureter were sutured. During the procedures, the blood supply for the ureter was protected. No hydronephrosis was detected after the operation in the five patients with ureteropelvic stenosis, and postoperative imaging did not show exudation around the anastomotic stoma, indicating the absence of anastomotic stenosis or leakage. Hitherto, only a few studies have used ICG in surgeries of patients with duplicate kidney and ureter [26]. Interestingly, for patient 12 in this study with duplicate kidneys and ureters in combination with lower renal calculi, percutaneous nephrolithotomy was performed before surgery, and a nephrostomy tube was implanted. The ICG was injected through the nephrostomy tube to display the segment of the lower ureteral stricture, and end-to-side anastomosis of the ureter with the lower renal pelvis. For the patients with primary ureteral tumors after renal transplantation, primary ureter, and sleeve bladdertomy were resected as it is challenging to treat the ureter and the transplanted renal artery. To the best of our knowledge, no previous studies have reported the use of ICG fluorescence imaging to guide the Da Vinci robot for the resection of an ipsilateral primary ureteral tumor after renal transplantation. Patient 14 in this study had severe peripheral tissue and organ adhesion due to renal transplantation surgery and inflammatory exudation of the tumor; also, the ureter was closely adjacent to the transplanted renal artery, and distinguishing and separating the renal artery is difficult. After ICG was injected through the ureteral cathedral, the fluorescence imaging system of the Da Vinci robot displayed the correlation between the ureter and surrounding organs and clarified the ureteral vessels and range of ureteral separation, thus simplifying the surgery and protecting the function of the transplanted kidney to a certain extent. ICG in combination with Da Vinci Xi robot for intraoperative fluorescence imaging promoted the procedures of key steps, especially in complex conditions and confusing anatomical structures, which reduced unnecessary surgical separations, promoted the blood supply for normal ureter, and shortened the operative time.

The operation duration and estimated blood loss during the operation did not significantly between the 14 patients and in previous studies [27]. Postoperative anastomotic stenosis or leakage is common in patients who have undergone ureteral and bladder reimplantation and pyeloureteroplasty [28,29,30]. Intraoperative imaging by ICG in combination with the Da Vinci robot increases the quality of anastomosis and prevents leakage or restenosis after anastomosis. After the anastomosis of the ureter to the bladder or renal pelvis, ICG was reinjected through the collecting system of the urinary tract; no ICG exudation or accumulation was detected, indicating the absence of anastomotic stenosis or leakage. Additionally, no ICG-related side effect was noted after the surgeries, which could be attributed to the method of injection. The D-J tube was removed in 13 patients at 3 months after the surgery, of which the hydronephrosis and ureterectasia improved substantially; strikingly, the renal function did not worsen further.

Nevertheless, the present study has several limitations. Firstly, the sample size of this study is small, and it is a case series study that lacks a control group. Secondly, although the ICG injection through the collecting system of the urinary tract is safe, this method is not included in the scope of ICG usage. Finally, the patients were only followed up for 3 months; hence, no ICG injection-related side effects were observed. Thus, it is unclear whether the relevant side effects would appear in a prolonged follow-up.

## 5. Conclusions

In the present study, ICG injection through the collecting system of the urinary tract was administered in combination with Da Vinci robot for intraoperative navigation in the complex upper urinary tract surgery. It compensated for the inadequacy of the tactile feedback of the surgical operating system and was capable of assisting operators to rapidly identify the stricture segment of the ureter and shortened the operative time. The measurement helps in assessing the anastomotic stenosis or leakage and also avoids the risk of systemic exposure to ICG.

## Figures and Tables

**Figure 1 jcm-12-01980-f001:**
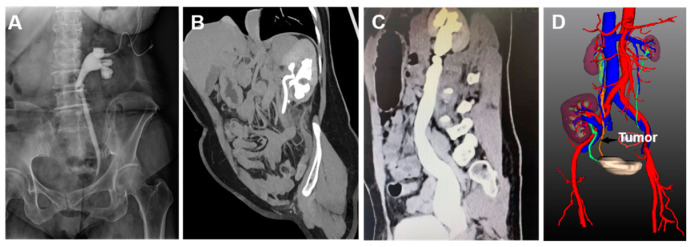
Images of patients. (**A**): Patient 1, KUB figure shows distal ureteral stricture and dilation of renal pelvis and ureter; a nephrostomy tube was implanted; (**B**): patient 12, multidimensional reconstruction CT image shows duplicate kidney and ureter; the ureter shows Y-shaped duplication; (**C**): patient 13, reconstructed CT image shows full-length dilation of the renal pelvis and ureter, as well as distal ureteral stricture; (**D**): patient 14 underwent renal transplantation and has a secondary tumor of the native ureter. CT three-dimensional reconstruction image shows that the native ureter is closely adjacent to the renal artery of the transplanted kidney.

**Figure 2 jcm-12-01980-f002:**
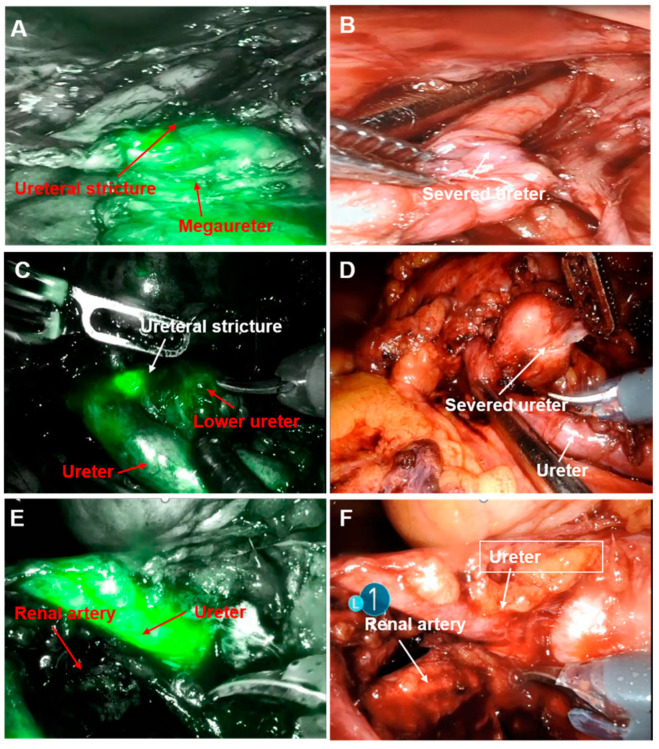
Fluorescence positioning of the ureter and the segment of the ureteral stricture. (**A**,**B**): Fluorescence imaging by Da Vinci robot in combination with white light imaging for the positioning of stricture of the lower ureter; (**C**,**D**): positioning of lower ureteral stricture in a patient with duplicate kidney and duplicate ureter by Da Vinci robot fluorescence imaging, after which the ureter was resected at the site of restructure; (**E**,**F**): native ureter by Da Vinci robot fluorescence imaging in the patient who underwent renal transplantation, and thus the spatial correlation between the ureter and renal artery was clarified.

**Table 1 jcm-12-01980-t001:** General characteristics of patients.

Case	Gender	Age (Years)	Disease Type	Lesion Side	Affected Side Renal Function (mL/min)
1	Female	49	Distal ureteral stricture	Left	20.2
2	Female	37	Distal ureteral stricture	Left	25.2
3	Female	36	Distal ureteral stricture	Right	24.1
4	Female	34	Ureteropelvic junction obstruction	Left	31.2
5	Female	46	Ureteropelvic junction obstruction	Right	29.1
6	Male	56	Ureteropelvic junction obstruction	Right	25.1
7	Male	52	Ureteropelvic junction obstruction	Left	36.3
8	Male	41	Ureteropelvic junction obstruction	Left	20.9
9	Female	32	Duplicate kidney and duplicate ureter	Left	29.8
10	Female	49	Duplicate kidney and duplicate ureter	Right	34.1
11	Male	47	Duplicate kidney and duplicate ureter	Right	41.2
12	Male	54	Duplicate kidney and duplicate ureter	Left	40.1
13	Male	31	Megaureter	Right	17.5
14	Female	56	Ipsilateral native ureteral tumor after renal transplantation	Right	-

**Table 2 jcm-12-01980-t002:** Intraoperative and postoperative information of 14 patients.

Case	Operation	ORT (min)	EBL (mL)	ETUS (min)	AS/AL	LOST	Renal Function 3 Months after the Operation (mL/min)
1	Ureter bladder replantation	98	50	21	No	7	37.4
2	Ureter bladder replantation	121	100	21	No	8	32.9
3	Ureter bladder replantation	171	100	18	No	7	41
4	Ureter bladder replantation	145	50	23	No	7	38
5	Ureter bladder replantation	134	50	20	No	8	37.3
6	Ureter bladder replantation	160	80	24	No	8	32.5
7	Ureter bladder replantation	167	50	20	No	7	41
8	Ureter bladder replantation	133	120	17	No	7	34
9	Repeated nephroureterectomy	112	50	25	No	9	45.3
10	Repeated nephroureterectomy	156	90	21	No	9	41.3
11	Repeated nephroureterectomy	143	50	19	No	8	47.1
12	Pyeloureteral anastomosis	173	60	18	No	10	42.4
13	Ureter bladder replantation	123	50	32	No	7	22.2
14	Native ureterectomy	169	140	43	-	10	-

ORT = operative time; EBL = estimated blood loss; ETUS = exposure time of ureteral stenosis; AS/AL = anastomotic stenosis or leakage; LOHS = length of hospital stay; exposure time of ureteral stenosis refers to the time from fluorescence imaging of ureteral stricture to surgical resection of the stricture segment.

## Data Availability

All data generated or analyzed during this study are included in this article.

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
