# Peer review of "Application of Indocyanine Green in Combination with Da Vinci Xi Robot in Surgeries on the Upper Urinary Tract: A Case Series Study"

_jcm, 2023, doi:10.3390/jcm12051980_

Round 1
Reviewer 1 Report
This report describe intraluminal application of ICG for visualization of ureteral stricture in robotic surgery utilizing DaVinci Xi.
Major problems
1. The discussion about advantage of ICG over conventional laparoscopic/robotic surgery, which relied exclusively upon white light image, is lacking. It is unclear whether these cases really needed ICG, or not. In reoperative case like Case 14, the post-transplant case, it may be difficult to identify the ureter, and such cases may benefit by ICG. The authors should provide such information for the remaining cases. In majority of the cases, the authors placed percutaneous nephrostomy for injecting ICG, an invasive procedures which may be unnecessary if there is no extensive adhesion. Were the patients informed about it? Conversely, in how many cases retrograde visualization was successful?
2. It is difficult to understand Figure 2. Schema or cartoons are required to clarify the location of the stricture site, in relation to surrounding structures.
3. The authors use terminology, if not wrong, but different from standard medical idioms. For example
‘primary ureter’ -> native ureter
‘giant (large) ureter’ -> megaureter
‘anterograde’ -> antegrade
‘renal pelvis fistulation’ -> nephrostomy
Minor comments
4. How did they determine the dilution of ICG?
5. Spelling and grammatical errors
LOST -> LOHS
cathedral -> catheter
Reviewer 2 Report
The authors present a new option for surgery for ureteral stricture.
1. Please delete Table 2 as there are two Tables.
2. Perioperative and postoperative outcomes, such as operative time, are written twice. Please correct.
Reviewer 3 Report
Dear authors, congratulations for your hard work. It is a surgeon's nightmare, when he agonizes to find the ureter, and the ureter simply isn't there! Your work points to the right direction. Nevertheless, as you elegantly mention, your paper has limitations and flaws if I may say so. Your study lacks control sample. It is mandatory to have control sample, in order to estimate any significance, and of course your sample is small, since it isn't omogenous, but you have dealt with various cases. Now regarding the short follow-up, it would have been better if you had evaluated your patients in 1 month, 3 months and 6 months period of time.
I hope that you will find my remarks useful, and I strongly urge you to keep up your good work.
Reviewer 4 Report
1-add a paragraph in introduction method and mention the possible adverse effects of ICG injection.
2-the title of your manuscript is "complex surgeries of upper urinary tract", the question is: what were the complexities of your UPJO patients?
3- the second paragraph of result section is repetitive, please rewrite it.
4- as you said in your manuscript, you inject ICG through collecting system for the first time, on what basis did you used this substance through collecting system? has an animal study been done in this regard, if yes please reference it
Round 2
Reviewer 1 Report
The article is appropriately revised
Reviewer 3 Report
Dear authors, keep up the good work, looking forward to read your updated work.
Reviewer 4 Report
dear author
your manuscript is proper for publication